# Radial or Focal Extracorporeal Shock Wave Therapy in Lateral Elbow Tendinopathy: A Real-Life Retrospective Study

**DOI:** 10.3390/ijerph20054371

**Published:** 2023-02-28

**Authors:** Raffaello Pellegrino, Angelo Di Iorio, Serena Filoni, Paolo Mondardini, Teresa Paolucci, Eleonora Sparvieri, Domiziano Tarantino, Antimo Moretti, Giovanni Iolascon

**Affiliations:** 1Department of Scientific Research, Campus Ludes, Off-Campus Semmelweis University, 6912 Lugano, Switzerland; 2Antalgic Mini-Invasive and Rehab-Outpatients Unit, Department of Innovative Technologies in Medicine & Dentistry, University “G. d’Annunzio”, 66100 Chieti, Italy; 3Padre Pio Foundation and Rehabilitation Centers, 71013 San Giovanni Rotondo, Italy; 4Department of Sport Science, Università di Bologna, 40100 Bologna, Italy; 5Physical Medicine and Rehabilitation, Department of Oral Medical Science and Biotechnology, University “G. d’Annunzio”, 66100 Chieti, Italy; 6Internal Medicine Unit, Ospedale S. Liberatore, 64100 Teramo, Italy; 7Department of Public Health, University Federico II of Naples, 80131 Naples, Italy; 8Department of Medical and Surgical Specialties and Dentistry, University of Campania “Luigi Vanvitelli”, 80138 Naples, Italy

**Keywords:** extracorporeal shock wave therapy, lateral elbow tendinopathy, muscle strength, gender effect, rehabilitation, minimally invasive

## Abstract

Lateral elbow tendinopathy (LET) is characterized by pain, poor muscle strength of the wrist ex-tensors, and disability. Among the conservative rehabilitative approaches, focal as well as radial extracorporeal shock wave therapy (ESWT), are considered effective in LET management. The objective of this study was to compare the safety and effectiveness of focal (fESWT) and radial (rESWT) in terms of LET symptoms and the strength of wrist extensors, taking into account potential gender differences. This is a retrospective longitudinal cohort study of patients with LET treated with ESWT that had received a clinical and functional evaluation, including visuo-analogic scale (VAS), muscle strength using an electronic dynamometer during Cozen’s test, and the patient-rated tennis elbow evaluation (PRTEE) questionnaire. Follow-ups were carried out weekly in four visits after enrollment, and at 8 and 12 weeks. During the follow-ups, the VAS score decreased in both treatments, even if patients receiving fESWT reported early pain relief compared to those treated with rESWT (time for treatment *p*-value < 0.001). Additionally, peak muscle strength increased independently of the device used, and again more rapidly in the fESWT group (time for treatment *p*-value < 0.001). In the stratified analysis for sex and for the type of ESWT, rESWT appears to be less effective in female participants in terms of mean muscle strength and PRTEE scores, without differences according to the type of device used. The rESWT group reported a higher rate of minor adverse events (i.e., discomfort, *p* = 0.03) compared to fESWT. Our data suggest that both fESWT and rESWT might be effective in improving LET symptoms, even if the higher rate of painful procedures were reported in patients treated with rESWT.

## 1. Introduction

Tennis elbow, also known as lateral elbow tendinopathy (LET), is characterized by chronic degeneration at the origin of the extensor carpi radialis brevis muscle on the lateral epicondyle of the humerus [1]. LET represents one of the most common tendinopathies of the upper extremities, with an annual incidence of 1–3% in the total population [1,2]. The prevalence of LET peaks between the ages of 30 and 60 years, and LET primarily affects the dominant arm, with symptoms appearing for a longer duration and with greater severity in women [3]. Tennis elbow is considered to be self-limiting; in 80% of patients, the symptoms resolve within six to twelve months [4]. It is usually caused by injury or overuse. The etiology of this disorder is related to a process that, in tendons damaged by repetitive microtrauma, leads to vascular proliferation and hyaline degeneration (angiofibroblastic hyperplasia) [1]. Several invasive and non-invasive treatment protocols have been proposed for LET management [5]. Conservative treatments include physical therapy, eccentric exercises, laser therapy, acupuncture, epicondylar elbow braces, and drug therapy including corticosteroid injections, botulinum toxin, autologous blood, and platelet-rich plasma [2].

Extracorporeal shock wave therapy (ESWT) is a noninvasive procedure considered safe and well tolerated by most of patients with LET [6], and the treatment of radial epicondylitis by ESWT has obtained FDA approval in the United States [7]. There are two different types of shock waves in clinic: focused shock wave therapy (fESWT) and radial shock wave therapy (rESWT). However, there are considerable differences between the two types of shock waves concerning, for example, the quality of the sound fields and the focus area. Focused shock waves are generated inside the applicator and then focused by a lens and transmitted into the tissue. The radial devices use compressed air or electro-magnetic forces to accelerate a ‘projectile’ in the device, which transfers its energy on impact and applies it to the tissue [8]. The two devices act in somewhat different ways; fESWT are concentrated on a restricted area of the body and can penetrate deep into the tissue, whereas rESWT acts with pressure at the skin surface and then diverges, even if it cannot be excluded that positive effect in the depth will also be reached [9]. An alternative hypothesis considers that shock waves relieve pain in insertional tendinopathy by hyperstimulation analgesia. An initial increase and subsequent long decrease in substance *p* released from the treated region could explain the initial pain during and shortly after shockwave treatment of tendon insertion and subsequent lasting pain relief [6]. Conflicting evidence is available about the effects of ESWT for the management of LET. In a meta-analysis, ESWT did not show clinically important improvement in pain reduction and grip strength. On the other hand, rESWT and a short follow-up seem to be related to better control of LET symptoms [10]. Recently, our group, in a retrospective study assessing an integrated approach ESWT plus hyaluronic acid, showed the role of gender in the response to the fESWT approach to rotator cuff tendinopathy; in those patients receiving fESWT alone, men reported higher benefits in terms of pain relief compared to women [11]. In a pilot experimental study comparing the two ESWT techniques, focusing on radial shock wave therapy efficiently controlled chronic pain complaints and restored muscle strength [12]. 

We hypothesize that focal and radial ESWT act with different modalities in the healing processes, and that using different energy intensity or contact pressure on the tissues can induce procedural pain or side effects (i.e., petechial bleeding) that could influence the patients’ compliance. Therefore, the main objective of this study was to compare the effectiveness and safety of fESWT and rESWT in terms of pain relief, strength recovery, and side effects, respectively, taking into account the role of gender in this context.

## 2. Methods

We performed a retrospective longitudinal cohort study including medical records of patients with painful LET referred to the Chiparo Physical Medicine and Rehabilitation outpatient in Lecce between January 2021 and December 2021. The study was planned at the Clinical Research Department of the Ludes Campus in Lugano (CH), and complies with the STROBE guidelines of the Consort Statement for the reporting of observational studies.

To be eligible, patients had to be between 30 and 65 years old with elbow pain intensity on the visuo-analogic scale (VAS) of at least 50, and had to be negative upon screening for other sources of pain. The diagnosis of LET had to be confirmed by US evaluation showing structural inhomogeneity and altered tendon thickness. 

The study was developed following the Good Clinical Practice (GCP) guidelines. It was conducted within the ethical principles outlined in the Declaration of Helsinki, and with the procedures defined by the ISO 9001-2015 standards for “Research and experimentation”. Written informed consent to provide information included in personal medical records was obtained from all participants. No Ethics Committee was required, since this is a real-life (retrospective observational) study that compares two medical devices that are routinely used in clinical practice.

### 2.1. Outcome Measures

At baseline, all participants completed a sociodemographic questionnaire and anamnestic interview regarding the presence of comorbidities, medication intake, and potential risk factors for adverse drug reactions. Patients underwent functional and ultrasound examination of the elbow. Follow-ups were carried out weakly in four visits after enrollment (W1-W4), and at 8 and 12 weeks (W8, W12). In addition, patients completed the VAS (0–100, where 0 = no pain) at every follow-up (Wbaseline-W12). However, at baseline, W4, W8, and W12 was assessed with the patient-rated tennis elbow evaluation (PRTEE) questionnaire [13], and a measurement of maximum force and mean force using an electronic dynamometer during contraction versus resistance of wrist extension with the elbow flexed at 90° (Cozen’s test) [14]. Participants performed a total of three maximal contraction trials lasting 5 s each, interspersed with 15 s of rest [14].

### 2.2. Extracorporeal Shock Wave Therapy

#### 2.2.1. Focal Extracorporeal Shock Wave Therapy

The focal group received three sessions of f-ESWT (at day 1, day 7, day 14), administered by the same operator. For each single therapeutic session with f-ESWT, 1500 pulses emitted by focused shock wave generator (piezoclast-EMS ESWT) with a range of energy flux density between 0.12–0.18 mj/mm^2^ and a frequency between 5 and 8 Hz, with 5 Hz for the first session and 8 Hz for the second and third sessions, respectively, were administered. The treatment was administered regularly, and titrated up to make sure it was always perceived as “strong but comfortable” during use. During the therapeutic procedure, patients were seated, and the affected elbow was positiond at 90° of flexion and in slight pronation. All impulses were transmitted in the epicondylar area, at the site of the insertion of the common extensor tendon. An ultrasound trasmission gel was placed between the applicator of ESWT and the skin.

#### 2.2.2. Radial Extracorporeal Shock Wave Therapy

The radial group received three sessions of r-ESWT (at day 1, day 7, day 14), administered by the same operator. For each single therapeutic session with r-ESWT, 1500 pulses emitted by radial shock wave generator (Dolorclast Master-EMS) with a range of pressure between 2.0–2.5 Bar and a frequency between 08 and 13 Hz, 8 Hz for the first session and 13 Hz for the second and third sessions, respectively, were administered. The treatment was administered regularly, and titrated up to make sure it was always perceived as “strong but comfortable” during use. During the therapeutic procedure, patients were seated, and the affected elbow was positiond at 90° of flexion and in slight pronation. All impulses were transmitted in the epicondylar area, at the site of the insertion of the common extensor tendon. An ultrasound trasmission gel was placed between the applicator of ESWT and the skin.

### 2.3. Sample Size

In subjects treated with ESWT, in the short-medium term (12 weeks), the reduction in the pain reported is evaluated to be 70% of the initial level [10]. We hypothesized a mean difference of 20% in the VAS between the two ESWT approaches at the follow-up, a population study standard deviation of 16, and an intraclass correlation coefficient ρ = 0.30, with an alpha error α = 0.05. Therefore, at least 36 subjects per group had to be enrolled to obtain an estimated power β > 90. Lastly, to account also for attrition (15%), we decided to enroll at least 7 more patients in the study for each study groups. The final overall sample size was 84 patients.

### 2.4. Statistical Analysis

Data were reported as mean ± standard error (S.E.) for continuous variables, and as an absolute number and percentage for dichotomous variables; differences between groups were assessed with analysis of variance and Chi-square test, respectively. All analyses were conducted with an intention to treat analysis. To assess variations in VAS score, pick muscle torque, mean muscle strength, PREE total score, self-reported pain subscale and specific and non-specific disability score, linear mixed models were applied [15]. The advantage of this approach is that it increases the precision of the estimate by using all available information concerning performance and, at the same time, allows for handling missing data and had a more powerful modelling of the analysis. Intercept and time had a random component for all the variables except PRTEE total score and its subscale. A different strategical approach was followed to analyze subscale and total score of PRTEE, due to the low variability of the scale and due to the fact that almost all the variance was linked to the “between subject” level and/or to the time level. Therefore, instead of modelling data with a random intercept and time, we use the repeated model option, considering therefore only the saturated model, with time, treatment and their interaction. In LMMs, the two treatments were considered with ES-F as the reference group. The times of the study were considered with W12 as the reference, as was the interaction between time and treatment. Moreover, to assess the effect of sex, stratified analysis was conducted for sex wherein female patients were the reference group, and only for this analysis was drop-out excluded. Data were analyzed with SAS software (rel. 9.4), and the *p*-value for differences was considered statistically significant at a value less than or equal to 0.05.

## 3. Results

In the study, 84 patients were enrolled; of those, 46 (54.8%) were females, with a mean age of 46.7 ± 8.4 years, and 50 (59.5%) reported pain at the right elbow, without statistically significant differences between study groups. A total of 24 (57.1%) were enrolled in the rESWT compared to 26 (61.9%) of the fESWT (*p*-value = 0.66). Almost 40% of the patients were white-collar workers, and 22 (26.2%) practiced sport (see also Appendix A, in which a comparison of this element is reported between the two treatments). The VAS score decreased over time in both groups; however, an estimated 11% (ICC) of the total variation in the self-reported pain level is attributable to differences between patients (Table 1, Model A). 

In Table 1, Model B reported the effect of time, where T12 is the reference. On average, at T12, the VAS score is equal to 24.1 ± 1.5, and for every follow-up, an increase of 8.8 ± 0.2 points (γ10) could be seen; 92% of the within-person variation in pain is associated with linear time (R^2^ ε). When the interaction between time and treatment is considered in the model, a multiplicative effect could be demonstrated, favoring fESWT (Table 1, Model C γ11 = −9.4 ± 0.4, *p*-value < 0.001). With the interaction term in the model, the ρ decreased from 0.84 to 0.38, meaning that a smaller fraction of the total variation is influenced from the higher starting level of pain. In Figure 1, the VAS score according to time, treatment and sex is reported; moreover, the estimates from LMM of the interaction “sex for time”, within the entire sample of patients included in the study, except those that dropped out, were also reported in the following groups: independently of treatment (Model A), patients that underwent ES-R (Model B), and lastly, patients that underwent ES-F (Model C). Female patients reported a higher VAS score in the entire population (Model A), at W4 (*p*-value = 0.002), W3 (*p*-value = 0.004), and W2 (*p*-value = 0.002) compared to W12. Almost all the differences between sex could be attributed to rESWT treatment, since no differences could be demonstrated in the VAS scores between sex in the fESWT group.

In both treatment groups, muscle pick torque increased over time; an estimated 23% (ICC) of the total variation in the strength is attributable to differences between patients (Table 2, Model A). 

On average, at T12, muscle pick torque was γ00 = 121.5 ± 1.7 Nw., and for every follow-up, a decrease in muscle strength could be found (γ 10 = −5.5 ± 0.3, *p*-value < 0.001, Table 2, Model B). When the interaction term between time for treatment is considered in the model, fESWT seem to be less effective compared to rESWT (γ01 = −11.2 ± 3.5, *p* < 0.001), and a multiplicative effect can be also demonstrated (γ11 = 1.3 ± 0.5, *p*-value < 0.001, Table 2, Model C). In Figure 2, variation in muscle pick torque according to time, treatment, and sex is reported. Additionally, in this case, muscle pick strength increased less in the female sex compared to male, but only at W4 in the Model A that considers the whole population of the study and in Model B that considers those patients that underwent rESWT; this is not the case for those patients treated with fESWT in Model C.

Muscle mean strength increased over time in both groups, but no multiplicative effect for the interaction between groups and time could be reported; moreover, no statistically significant differences could be seen between sex (Appendix A). On average, at T12, the mean muscle strength score is equal to 112.8 ± 1.7 Newton, and for every follow-up, a decrease of −6.2 ± 0.2 Newton could be seen (*p* < 0.001). 

To analyze the total score, pain subscale, and disability subscale of the PRTEE in the variation across times, we could not use intercept and time as random, but had instead to use the repeated measures option, because all the variability was linked to the “between subject” level and/or to the time level. In Figure 3A–C, the variation across times of the subscales and in the PRTEE total score are reported; at baseline, statistically significant differences between treatments could not be found. At W4, both study groups demonstrated a significant reduction in self-reported pain, in the disability subscale, and in total PRTEE scores. In the population study independent of treatments, the PRTEE total score decreased from baseline to W4 (−39.4 ± 1.7; *p*-value < 0.001) and to W12 (−52.5 ± 2.2; *p*-value < 0.001). At W4, the ES-R group reported higher PRTEE scores (5.5 ± 2.4; *p*-value = 0.02) compared to ES-F; those differences were no more statistically significant at W8 and W12. Excluding from the analysis those patients that interrupted the treatment, the female sex at W4 (4.8 ± 1.3; *p*-value < 0.001) and W8 (2.7 ± 1.2; *p*-value = 0.03) responded less to the ES-R approach in their PRTEE total scores (Figure 3A). A similar figure could be seen in pain subscales (Figure 3B), and in the disability subscale (Figure 3C) of the PRTEE, but only for those female patients treated with ES-R, and not for those treated with ES-F.

Fifteen patients out of 84 (17.9%) developed an adverse event (severe pain, hemorrhagic suffusion, or refused the treatment for local discomfort); of those, 11 (26.2%) were in the rESWT group and four in the fESWT group (*p*-value = 0.03). Among those 15 patients, ten out of the 84 enrolled dropped out and shifted towards another therapeutical rehab approach. Among them, six (14.3%) were in the rESWT group, and four (9.5%) were in the fESWT treatment; three were males, and seven (15.2%) were females (*p*-value = 0.16).

## 4. Discussion

Our findings suggest that both fESWT and rESWT might be effective in the treatment of LET. In four weeks, both treatments can reduce self-reported pain level, increase peak and mean muscle strength and reduce the PRTEE-score. As time passed, both treatments resulted in reduced scores in the VAS, although focal device treatment was more effective early and in the female sex.

Even if fESWT and rESWT differ in the physical characteristics of waves and in the treatment protocol parameters (i.e., energy density, impulses, frequency, number of sessions, local or diffuse application method, and frequency of treatments), these interventions share the ability to promote tissue healing, to induce the release analgesic mediators and to relieve pain [9,16]. The therapeutic efficacy of fESWT and rESWT in the LET treatment, with respect to the strength dysfunction period, was assessed by Stania et al.; they found that both approaches are equally effective for acute and chronic LET [12], and therefore the therapeutic effect does not depend on the duration of pain symptoms. Similarly, Król et al. conducted a pilot RCT to assess the efficacy of focal compared to radial ESWT; both treatments provided a gradual reduction of pain and an improvement in the strength of the affected extremity [17]. Conversely, a Cochrane meta-analysis evaluating the effectiveness and safety of ESWT for LET treatment found that ESWT provides little or no benefit in terms of pain relief [18]; however, some study criticisms need to be considered. For example, in the analysis, acute and chronic LET were considered, but there was no differentiation between studies that used focal or radial ESWT. Lastly, there are lack of sufficient data to draw powerful comparisons and analysis. Similar results emerged from a recent meta-analysis conducted by Karanasios et al., reporting low-to-moderate certainty of evidence that ESWT, without differentiation between the focal or radial modality, has no clinical benefits compared with placebo in LET treatment [19].

Therefore, no clear evidence of superiority of one of the two type of ESWT could be derived from analysis of the literature [10]. In our observational retrospective analysis, fESWT was more effective compared to rESWT, with a clear female gender difference. 

The rationale for explaining better results obtained through fESWT could be that fESWT is concentrated on a restricted area and can penetrate deeper in the body com-pared to rESWT [9]. Moreover the focused devices use a different energy profile with a rapid rise and decrease of the wave, and higher maximum pressure and longer pulse du-ration compared rESWT [20]. Therefore, all these impulse characteristics could play a role in the different effectiveness of ESWT for managing LET patients. On the other hand, the role of procedural and post-procedural pain could be hypothesized. During ESWT, pain and discomfort may be reported by patients [21], inducing the operator to reduce the intensity of the shockwave. In our data, this hypothesis could be partially supported by the higher rate of adverse events reported by the rESWT treatment group, which was even more evident in the female sex. Moreover, this hypothesis could also explain the lower benefits of rESWT in the management of pain and strength in LET, considering that female patients treated with rESWT need a longer time for pain relief and strength recovery. In our work, the study design could influence the distribution of adverse effects between groups. 

Even if the most frequent intra-intervention complaint was pain in the shock wave application area, other adverse effects of ESWT may include transient skin reddening, small areas of subcutaneous extravasation resulting from small vessel injury, bruises, local swelling and numbness of the area [22]. Looking at the results of our study, both ESWT treatments were found to be well tolerated and safe, even if radial treatment showed a higher rate of minor adverse events compared to focal treatment; this is probably related to the different wave profile in the two ESWT treatments [19]. 

While the two ESWT approaches were shown to be effective in pain control and strength recovery, conversely, focal and radial ESWT seemed less effective in improving the PRTEE subscale than the total score. From our point of view, the computational way that the PRTEE subscales and total score are calculated could influence the low variability of the scale, linking almost all of the variance to the “between-subjects” level [13]. This could mean that in our study population, PRTEE intercepted differences in the quality of life (in the perception of disability) between patients. In other words, the same pain score could be reported with different disability levels. Alternatively, the PRTEE subscale considered pain and disability in specific and nonspecific activities; those two components of disability could influence the variability of the scale. As matter of fact, the VAS score and PRTEE pain subscale showed a strong correlation only at the baseline, independently of the study group. On the contrary, disability subscales showed only a modest, and not statistically significant, correlation with VAS score and muscle strength among the different times of the study (Appendix A).

The most important limitation to be considered in our work is the study design, as previously stated. Additional observational retrospective analysis could help to elucidate scientific questions, especially if data were drawn from a real-life clinical approach. Moreover, a sham control group or exercise-only treatment group could be helpful in disentangling the question of the effectiveness of the ESWT in LET treatment. However, the results of previous studies showed that either f-ESWT or r-ESWT had an advantage over a placebo in terms of pain relief and knee function improvement in those patients [4]. Another limitation is the lack of assessment of analgesic drug use in the follow-up, which potentially could have modulated the results of our analysis. Lastly, our sample size study was conducted to assess potential differences between the two ESWT approaches in pain management. Therefore, the small number of side effects registered in the study, on one hand highlighted and confirmed that ESWTs are safe, but on the other hand could suggest that our study is underpowered to this outcome. Therefore, upcoming studies need to be sampled and account also for side effect analysis.

## 5. Conclusions

Keeping in mind the limitations due to the observational retrospective design of our study and the lack of sham control group, ESWT seems to be a valid alternative to the classical exercise approach in the treatment of pain, disability, and muscle impairment in LET; this is the case even if in the female sex, rESWT seems to be less effective and requires more time to achieve pain relief and functional recovery.

## Figures and Tables

**Figure 1 ijerph-20-04371-f001:**
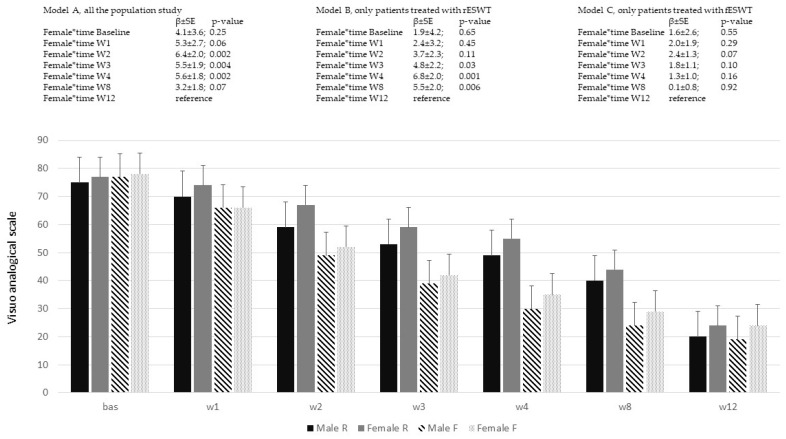
Visuo-analogic scale variation across time of the study and sex. In the models, the estimates (β ± SE) and the *p*-value derived from linear mixed model analyses were reported. Model A: whole population study; Model B: patients treated with rESWT; Model C: patients treated with fESWT. The histogram identifies Male R and Female R, patients treated with radial ESWT; Male F and Female F, patients treated with focal ESWT.

**Figure 2 ijerph-20-04371-f002:**
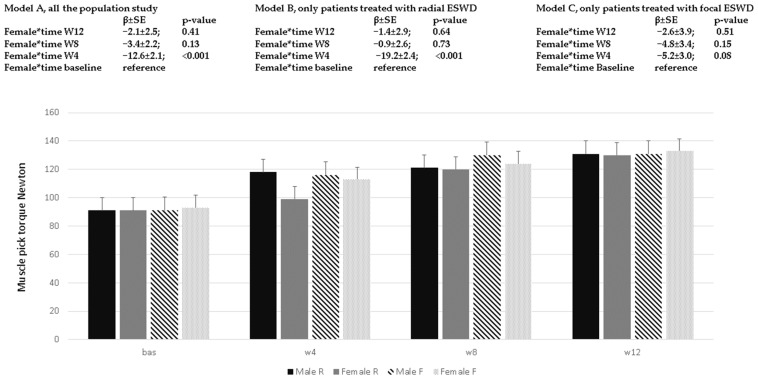
Muscle pick torque (Newton) variation across time of the study and sex. In the different models, estimates (β ± SE) and *p*-values derived from linear mixed model analyses were reported. Model A: all the population study; Model B: patients treated with rESWT; Model C: patients treated with fESWT. The histogram identifies Male R and Female R, patients treated with radial ESWT; Male F and Female F, patients treated with focal ESWT.

**Figure 3 ijerph-20-04371-f003:**
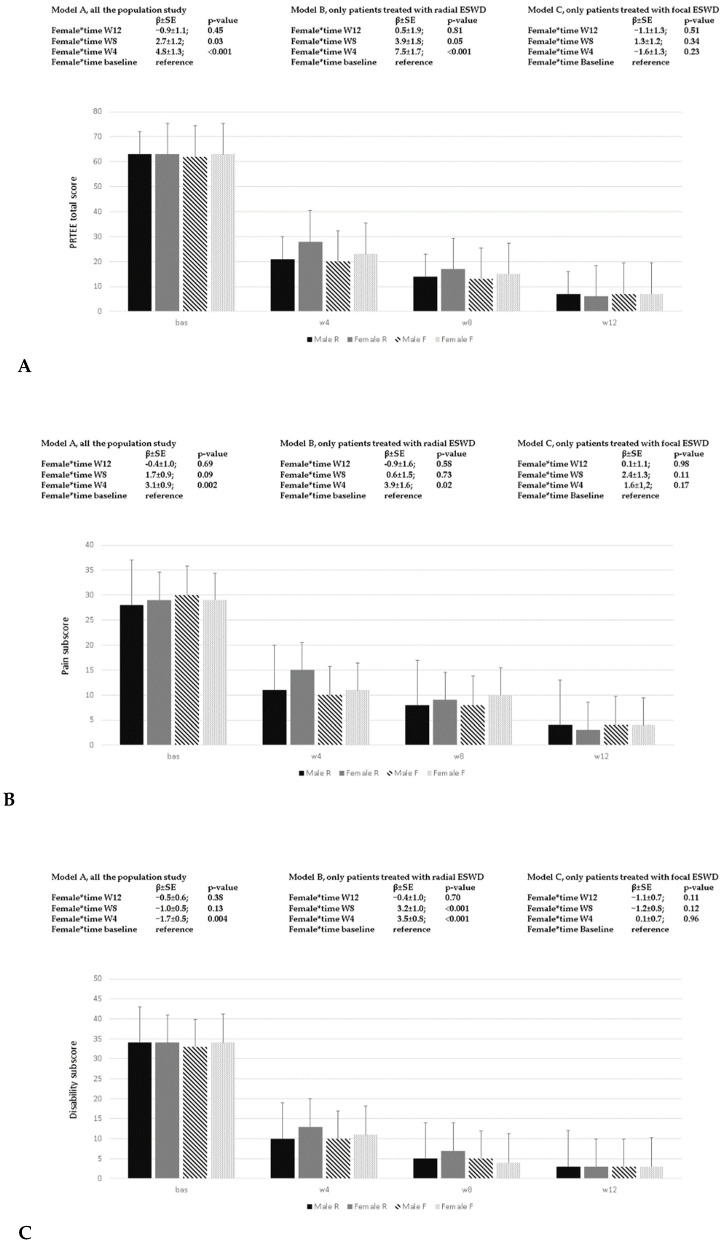
Patient-rated tennis elbow evaluation (PRTEE) questionnaire. In (**A**–**C**), the variation in the total score, pain subscale score, and disability subscale score according to sex and type of ESWT were reported. In the different models, estimates (β ± SE) and *p*-values derived from linear mixed model analyses were reported. Model A: whole population study; Model B: patients treated with rESWT; Model C: patients treated with fESWT. The histogram identifies Male R and Female R, patients treated with radial ESWT; Male F and Female F, patients treated with focal ESWT.

**Table 1 ijerph-20-04371-t001:** Linear mixed model: variation of visuo-analogic scale during the follow-up according to treatment. The last follow-up (T12) was the reference for the comparison among the times of the study. Model A: unconditional means model; Model B: unconditional growth model; Model C: saturated person-level model.

			Model A	Model B	Model C
Initial status	Intercept	γ _00_	50.2 ± 1.2 ***	24.1 ± 1.5 ***	−0.7 ± 1.9
	Treatment	γ _01_			50.9 ± 2.7 ***
Rate of change	Intercept (time)	γ _10_		8.8 ± 0.2 ***	13.3 ± 0.3 ***
	Treatment * Time	γ _11_			−9.4 ± 0.4 ***
Level 1	Within person	δ ^2^ _e_	425.7 ± 26.8 ***	35.3 ± 2.4 ***	16.1 ± 1.1 ***
Level 2	In initial status	δ ^2^_0_	52.9 ± 18.1 ***	95.1 ± 16.5 ***	85.3 ± 19.3 ***
	In rate of change	δ ^2^_1_		1.4 ± 0.3 ***	6.6 ± 1.2 ***
	Covariance	δ _01_		−1.3 ± 1.5	−9.7 ± 5.3
	ICC		0.11	0.73	0.84
		ρ		0.84	0.38
		R^2^ _y,y1_		0.66	0.57
		R ^2^ _e_		0.92	0.92
		R ^2^_0_			0.11
		AIC	5277	4152	3909

*** *p*-value < 0.001.

**Table 2 ijerph-20-04371-t002:** Linear mixed model: variation of pick muscle torque (Newton) during the follow-up according to treatment. The last follow-up (T12) was the reference for the comparison among time of the study. Model A: unconditional means model; Model B: unconditional growth model; Model C: saturated person-level model.

			Model A	Model B	Model C
Initial status	Intercept	γ _00_	11.5 ± 1.4 ***	121.5 ± 1.7 ***	126.8 ± 2.5 ***
	Treatment	γ _01_			−11.2 ± 3.5 ***
Rate of change	Intercept (time)	γ _10_		−5.5 ± 0.3 ***	−6.1 ± 0.4 ***
	Treatment * Time	γ _11_			1.3 ± 0.5 ***
Level 1	Within person	δ ^2^ _e_	316.9 ± 28.2 ***	53.9 ± 6.1 ***	51.1 ± 5.9 ***
Level 2	In initial status	δ ^2^_0_	93.2 ± 27.7 ***	72.3 ± 17.6 ***	70.5 ± 17.1 ***
	In rate of change	δ ^2^_1_		0.9 ± 0.3 ***	1.1 ± 0.3 ***
	Covariance	δ _01_		4.9 ± 1.6 ***	4.5 ± 1.6 ***
	ICC		0.23	0.58	0.58
		ρ		0.84	0.38
		R^2^ _y,y1_		0.44	0.45
		R ^2^ _e_		0.83	0.83
		R ^2^_0_			0.04
		AIC	2950	2593	2525

*** *p*-value < 0.001.

## Data Availability

The datasets used and/or analyzed during the current study are available from the corresponding author on reasonable request.

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
