# Peer review of "Radial or Focal Extracorporeal Shock Wave Therapy in Lateral Elbow Tendinopathy: A Real-Life Retrospective Study"

_ijerph, 2023, doi:10.3390/ijerph20054371_

Round 1

Reviewer 1 Report

Dear Authors,

The review of your manuscript has prompted the following comments:

2. The descriptive statistics on the patients you provided are incomplete.

2.1. Which of the VAS scales did you actually use?

2.2.1. The correct unit of Energy Flux Density for Focused Shock Wave should be stated.

2.2.1. and 2.2.2. How did you change the frequency of focused and radial shockwave during subsequent treatments? The description the experiment is insufficient to recreate it.

2.2.1. and 2.2.2. Did you use a coupling agent during treatments?

2.3. Why was the sample size calculated for pain only, disregarding the results of the PRTEE questionnaire and the mean and maximum force of the wrist extensors?

4. The trial by Król et al. [17] has demonstrated that the therapeutic effect of focused and radial shockwaves is similar.

Why do you use greater penetrability of focused shockwave compared with radial shockwave to explain its higher efficacy, when the treated site (the attachment of the extensor carpi radialis brevis to the lateral epicondyle) is very close to the skin’s surface?

 Transient skin reddening following the application of focused shockwave is indeed described in the literature as an undesirable effect, but the claim is hard to accept – it is a natural physiological reaction of tissue to a strong mechanical stimulus.

 I would expect an explanation of why more side effects were observed in the focused shockwave group compared with the radial shockwave group.

You should also explain how focused shockwave and radial shockwave reduce pain and improve strength and function of patients with lateral elbow tendinopathy.

Author Response

We have to thank this reviewer for the suggestions and the help

  1. The descriptive statistics on the patients you provided are incomplete.

- We apology but we are not sure to understand the suggestion. We reassessed the statistical approach and we believe that in the statistical-analysis-section was reported all the methods applied, with a large description of the models used; moreover, we have imagined that you could refer to data presentation, but we have considered and attached to the submission a supplementary Table 1, where are reported the characteristics of the study population and the differences between the two study groups. We suppose that this reviewer could not find the study design, but it was described at the beginning of the Methods section, it was an observational retrospective study. Lastly, if this reviewer would mean that is lacking a paragraph that describe how many patients were enrolled, we would like to explain that it was a choice, to avoid redundancy of data presentation. As matter of fact, the sample used was reported in the sample size analysis description, and in the first paragraph of the results. We hope, we have answered to Your doubt.

2.1. Which of the VAS scales did you actually use?

- Accordingly with your request we specify in the text the assessment used.

2.2.1. The correct unit of Energy Flux Density for Focused Shock Wave should be stated.

- We highlighted in yellow for both ESWT the reference-unit.

2.2.1. and 2.2.2. How did you change the frequency of focused and radial shockwave during subsequent treatments? The description the experiment is insufficient to recreate it.

- We have to thank this reviewer to point out this criticism, accordingly, we amended the text: (lines 127-131, and lines 140-141).

2.2.1. and 2.2.2. Did you use a coupling agent during treatments?

- As we stated in the limitations section, no “sham control group or exercise only treatment group” were included in the study design. Our study design was aimed to compare only the efficacy and effectiveness of both ESWT, therefore no rehab-exercise was scheduled for those patients. Obviously due to non “experimental” design of the study, patients were allowed to use pain-killer drugs; this data was assessed at baseline but not in the follow-ups.

We consider this as a limitation and accordingly, we insert a phrase in the opportune section (lines 341-343);

  • Moreover, since we are not sure about the interpretation of your observation, we also insert in the text, that an ultrasound transmission gel was used (lines 134 and lines 146).

2.3. Why was the sample size calculated for pain only, disregarding the results of the PRTEE questionnaire and the mean and maximum force of the wrist extensors?

- This is the methodological approach. The sample size needs to be calculated based on the main outcome, in this context we would like to verify the efficacy in term of pain control of ESWT. Therefore, our main outcome was the VAS variation in the study; accordingly, we calculated the sample on this score variation. From our point of view, the pain reduction is pivotal in the model, and only in a cascading effect, could determine an increase in performance and strength. 

+4. The trial by Król et al. [17] has demonstrated that the therapeutic effect of focused and radial

shockwaves are similar.

  • We appreciate a lot your observation, an accordingly we erase from the text: , but fESWT seem to be more effective compared to rESWT.

Why do you use greater penetrability of focused shockwave compared with radial shockwave to explain its higher efficacy, when the treated site (the attachment of the extensor carpi radialis brevis to the lateral epicondyle) is very close to the skin’s surface?

Accordingly, to your suggestion, in the hypothesis section we insert a phrase that could explain the reason (lines 71-75); moreover, from lines (306-319) we exposed our hypothesis that could be synthetized as:

  • fESWT are concentrated on a restricted area and can penetrate deeper (even if in LET-treatment could be plausible only partially);
  • a different energy profile with a rapid rise and decrease of the wave, compared to maximum pressure and longer pulse duration in rESWT;
  • procedural and post-procedural pain in ESWT (that is more frequent from our results in radial ESWT).

Transient skin reddening following the application of focused shockwave is indeed described in the literature as an undesirable effect, but the claim is hard to accept – it is a natural physiological reaction of tissue to a strong mechanical stimulus.

  • We agree with this reviewer, but as you stated the literature define “skin reddening” as an undesirable effect, and we could not have a different approach.

I would expect an explanation of why more side effects were observed in the focused shockwave group compared with the radial shockwave group.

  • Accordingly to your suggestion, we formulate our hypothesis (Lines 325-326)

You should also explain how focused shockwave and radial shockwave reduce pain and improve strength and function of patients with lateral elbow tendinopathy.

  • Accordingly to your suggestion, we formulate our hypothesis (Lines 293-294)

Reviewer 2 Report

Thank you for giving me the opportunity to review this manuscript. The concept of the study is interesting and the methodology is clear. 

I only have minor suggestions:

(a) please report participants' training or athletic experience. In addition, please report the age (mean/sd) of the male participants.

(b) Given the  observational retrospective, study design, and the absence of a control group, results should be interpreted with caution. I feel that the auhors should reshape their conclusions to fit the limitations of the study.

(c) Why muscle strength increased overtime in both groups? Please, clarify. 

(d) Please, correct the references list according to the journal requirements 

Author Response

Thank to this reviewer for the suggestions and the help

I only have minor suggestions:

  • please report participants' training or athletic experience. In addition, please report the age (mean/sd) of the male participants.

- to avoid redundancy of data presentation we only reported in the text few synthetic data, and as a supplementary material readers could find any specific related to the two study groups.

- Practiced sport 9 vs 13 patients (p=0.32) see supplementary Table 1;

- We did not find any statistically significant differences for age and sex (same supplementary table 1), but for your eyes only we compared age between sex also according to treatment group (via linear regression analysis).

Male: 45.2±7.5 vs Female: 47.8±8.9

CACO

1

-0.79380

1.83104

-0.43

0.6658

SEX

1

2.66513

1.83940

1.45

0.1512

  • Given the observational retrospective, study design, and the absence of a control group, results should be interpreted with caution. I feel that the auhors should reshape their conclusions to fit the limitations of the study.

- Accordingly to your suggestion we modify the conclusion.(lines 357-359).  

(c) Why muscle strength increased overtime in both groups? Please, clarify. 

- we insert in the introduction (hypothesis formulation) a phrase that we hope could clarify our hypothesis.

(d) Please, correct the references list according to the journal requirements 

We apology for the mistake and accordingly set the reference list.
